# Identifying Behavioural Traits and Underlying Personality Dimensions in Domestic Ferrets (*Mustela putorius furo*)

**DOI:** 10.3390/ani11082173

**Published:** 2021-07-22

**Authors:** Sarah Talbot, Rafael Freire, Skye Wassens

**Affiliations:** 1School of Agricultural, Environmental and Veterinary Sciences, Charles Sturt University, LMB 588, Wagga Wagga, NSW 2678, Australia; rfreire@csu.edu.au; 2Institute of Land Water and Society, Charles Sturt University, Albury, NSW 2640, Australia; swassens@csu.edu.au

**Keywords:** domestic ferrets, animal personality, behaviour, welfare

## Abstract

**Simple Summary:**

Personality research is lacking in the domestic ferret (*Mustela putorius furo*). The fundamental objective of this study was thus to investigate the underlying structure of personality in ferrets. Owner-reported trait ratings were collected using an online questionnaire, which entailed ferret owners rating several traits (adjectives) on a Likert scale, according to how well they applied to their ferret(s). Four key personality dimensions emerged and were labelled Extraversion, Sociability, Attentiveness, and Neuroticism, according to the types of traits they were composed of. Certain characteristics were found to influence personality: male ferrets were rated as more sociable than females, whereas females were rated as more attentive and neurotic. Extraversion generally decreases with age, and de-sexing was insignificant across all personality dimensions. These results are beneficial for informing the discussion concerning the evolutionary significance of animal personality as well as providing aid in positive welfare management, that is, catering to the needs of individuals rather than groups.

**Abstract:**

The aim of this study was to examine the personality structure of domestic ferrets (*Mustela putorius furo*) by using owner-based reporting of personality traits. A total of 743 ferret owners participated in an online questionnaire, with a total of 1029 ferrets being assessed. Respondents rated 67 adjectives based on their ferret(s) behavioural traits and personality. Principal component analysis (PCA) of these trait ratings identified four underlying personality dimensions, which accounted for 47.1% of the total variance. These were labelled according to the traits that they encompass: Extraversion, Sociability, Attentiveness, and Neuroticism. Details about ferret sex, de-sexing status, age, and coat colour were also sought, and General Linear Mixed Models were used to test the main effects of these characteristics on the personality dimensions. It was found that sex (*p* < 0.01) and age (*p* < 0.001) significantly influenced certain personality components, whereas de-sexing did not. Sociability, Attentiveness, and Neuroticism were found to differ based on sex, whereby males were rated as more sociable than females, but females were rated higher on the Attentiveness and Neuroticism subscales. Finally, Extraversion was found to generally decrease with age. We can use the findings of this study to make cross-species comparisons and further inform the discussion regarding the adaptive relevance of animal personality. Identifying differences in personality types can improve welfare by using this information to satisfy the different needs of individuals.

## 1. Introduction

Animal personality describes a distinctive pattern of behaviour within individuals and is stable in different situations and across time [1,2,3,4,5,6]. Personality can influence an individuals’ response to threats, new situations, and to conspecifics, and by making cross-species comparisons of personality dimensions, animal personality research aids in determining the functional significance of certain behavioural traits [7].

There are various different instruments, labels, and conceptualisations used to define personality, each with differing emphasis and interpretation [2]. The trait-rating method, such as that used to develop the Big Five Model of Human Personality [8], is the most popular tool for assessing personality and involves animal carers (i.e., animal owners, zookeepers, or trainers) using their extensive knowledge of each particular individual to rate the animal on a continuum of a certain trait (i.e., 1[timid]–5[bold]) [9]. Given their subjective nature, the ratings or scores of these traits are often referred to as subjective ratings (SRs).

Factor analysis of trait ratings provided by animal owners often reveals underlying relationships or constructs known as personality factors or dimensions within a population and can be used to describe the range of behavioural variation in a cluster of similar responses [10].

The practical applications of animal personality research are numerous. It can aid in providing personality-specific enrichment, improve compatibility between individuals in social groups, thus enhancing the physical safety of the group as well as the psychological welfare of individuals [11]. Regarding companion animals specifically, one useful application is for animal shelters to use animal traits to match prospective owners to suitable individuals, as well as to find appropriate carers for high care animals. This would improve adoption success as well as providing animals most in need of attention the care that they require. For example, the rate of successful adoption of dogs [12] and cats [13] has been found to increase with the use of such behavioural assessments [12].

Despite the ever-increasing body of animal personality literature, research investigating domestic ferret personality is still lacking. One integrative study [14] examined personality across six domestic species, of which ferrets were one, by using a rating instrument composed of human traits alone. This was due to the assessment tool being used across all study species, and human adjectives had already been subjected to substantial psychometric scrutiny during the development of the FFM [15]. Although this method allows for direct comparisons to be made across multiple domestic species, the trade-off was that species-specific traits were not included in the adjective list, meaning personality dimensions potentially unique to those species may not have been detected [4]. Furthermore, while the sample of ferrets used in this study was considered normative (*n* = 126), future research using more individuals would help to ensure the results are representative of the general population, as well as potentially capturing species-specific personality dimensions.

Ferrets have only been domesticated and lived alongside humans for approximately 2000–3000 years [16], compared to other companion animals such as dogs and cats, which have been domesticated for over 10,000 years [17,18,19,20]. Thus, it is likely that ferrets still share several behavioural traits with their wild counterparts and that we have not yet seen their complete evolutionary potential as a domesticated species. The relatively short domestication period of ferrets means that personality research in this species can potentially provide an insight into how personality evolves by making comparisons with other, more domesticated species.

The aim of this research was to investigate personality in the domestic ferret by identifying underlying trends in behavioural traits (personality dimensions). This was achieved by conducting factor analysis on trait ratings provided by owners. We also explored whether certain physical characteristics of ferrets were correlated with behavioural differences in individuals.

## 2. Materials and Methods

A broad list of adjectives that potentially describe domestic ferret personality and behavioural traits was developed by reviewing several human and animal behaviour and personality papers as well as behaviour or personality assessment tools developed previously [4,21,22]. Any words that could potentially describe traits in ferrets were selected by the researcher. This initial list of adjectives consisted of 232 words.

In order to reduce the number of adjectives on the list, a group of 10 experts was recruited. The group was comprised of experienced ferret owners (owned one or more ferrets for at least 10 years), as well as members of ferret rescue organisations, foster carers, and individuals who also used ferrets as hunting animals. The group participants were asked to rate each adjective on the list using a three-point scale (0 = does not apply to ferrets, 1 = possibly applies to ferrets, 2 = does apply to ferrets) to determine how well each adjective applies to ferret traits in general. The word list was then refined according to the experts’ responses, which involved eliminating adjectives that were considered to be ambiguous or to have limited variability. That is, if 80% or more of the experts agreed a particular adjective applies to ferrets, giving it a score of 2, or if 80% or more disagreed by giving an adjective a score of 0, then the adjective was excluded. This step was necessary to ensure that the words on the list discriminate between individuals; we were not interested in traits that the majority of the ferret population are observed to have [23]. The data were also descriptively analysed, and any adjectives identified as being ambiguous or confusing by at least one participant were removed. This resulted in the elimination of 165 adjectives, thus leaving a list of 67 words to describe the variability in ferret behaviour and personality.

With the collaboration of another researcher studying domestic ferrets (Dr. Marsinah Reijgwart of Utrecht University, The Netherlands), a questionnaire was then developed using the online survey tool, SurveyMonkey^®^, in both English and Dutch. The questionnaire was advertised through social media and welfare groups and was made available for six weeks (May–June, 2015) in countries where either English or Dutch was the language predominantly spoken. Any required translation between English and Dutch was conducted by Dr. Reijgwart. Informed consent was obtained from all respondents prior to commencing the questionnaire, whereby participants were required to read, and indicate that they understood, the information statement provided in order to progress to the questionnaire. The first part of the questionnaire comprised a combination of open- and closed-end and multiple-choice questions based on individual ferret characteristics (i.e., sex, de-sexed or intact, age, and coat colour). The next section focused on recognising pain in ferrets (Marsinah Reijgwart’s research). The final part of the questionnaire requested that ferret owners to rate the list of 67 adjectives on a 5-point scale (0 = not characteristic of my ferret, 1 = somewhat characteristic of my ferret, 2 = moderately characteristic of my ferret, 3 = really characteristic of my ferret, and 4 = word not applicable/unsure of how well this word describes my ferret). These trait ratings shall be referred to as subjective ratings (SRs). Furthermore, respondents had the choice to remain anonymous; however, there was a section at the conclusion of the questionnaire where respondents were able to indicate if they were interested in participating in further ferret-related research. This required respondents to provide their names and contact details. Approval was obtained from the Charles Sturt University, School of Animal and Veterinary Sciences (SAVS) Human Research Ethics Committee (#416/2014/03).

### Statistical Analysis

The initial output from a factor analysis of the data was checked to ensure the assumptions for principal component analysis (PCA) were met. The Kaiser-Meyer-Olkin Measure of Sampling Adequacy (KMO) value exceeded 0.6 (KMO = 0.9) [23,24], Bartlett’s Test of Sphericity value was significant (*p* < 0.01) [25], meaning PCA was appropriate for the data.

The rating scores from the questionnaire were subjected to principal component analysis (PCA) using SPSS version 20.0 [26]. To determine the underlying personality dimensions and how many should be retained, several guidelines [27,28] were reviewed and followed. This involved examination of the eigenvalues table (total variance explained table) and the scree plot [29], as well as conducting a parallel analysis using the statistical program Monte Carlo [30]. The eigenvalues table showed 11 components with eigenvalues exceeding one and accounting for 57.2% of the variance; however, the scree plot only showed a break after the fifth component, suggesting this instead should be the number of factors retained. Finally, the parallel analysis produced six factors with eigenvalues higher than the corresponding values for the same size (67 variables × 1029 respondents) randomly generated data matrix [31]. As a result, the four-, five-, and six-component models were investigated and interpreted using Varimax rotation.

For a large sample size (>100), it is generally recommended that the minimum cut-off for factor loadings is 0.4 [27,32]. Thus, after removing traits that did not load strongly (<0.4) on any of the components, or those that loaded strongly (>0.4) on multiple components, a four-factor model was deemed to be the best fit for the data and had the highest internal consistency between components.

Cronbach’s alpha (a measure used to determine internal consistency for which the value should be more than 0.7) [32] was calculated for each component. The SRs were then used to generate subscale scores for each ferret, and these were used to analyse the effects of ferret characteristics.

The assumption of normality was satisfied by examining histograms, residual plots, and Q-Q plots. GLMMs were thus used, followed by Tukey’s honestly significant difference (HSD) tests, to investigate the main effects of ferret characteristics on personality subscale scores. Lastly, inter-correlations between personality dimensions were analysed using Pearson’s correlations coefficients to test the hypothesis that the personality dimensions are significantly related to one another (i.e., the coefficients differ significantly from zero).

## 3. Results

There were 743 respondents who completed the questionnaire, with a total of 1029 ferrets assessed. Within this sample population, 424 (41.2%) ferrets were female, and 603 (58.6%) were male. It was more common for ferrets to be de-sexed (*n* = 894; 86.9%) than entire (*n* = 130; 12.6). The age group with the most ferrets was “2–6 years old” with 541 (52.9%) ferrets falling into this category, while the smallest age group was “more than six years old”, which consisted of only 92 ferrets (9.0%) (Table 1).

The four factors extracted from the data explained a total of 47.1% of the variance in the data and had a simple structure as recommended [33]. Table 2 illustrates the component loadings for each trait (and those traits removed from the model), the eigenvalues for each factor, and the percentage of variance explained by each component, and Cronbach’s alpha calculations. Two of the components (Sociability and Neuroticism) were just below the generally accepted Cronbach’s alpha value threshold of 0.7; however, this was not concerning given the large size used [34]. Component 1 loaded strongly on behaviours such as enthusiastic, playful, outgoing, and extraverted and was thus named Extraversion. The second component was labelled Sociability as it included items such as friendly, submissive, tame, and affiliative. Behaviours reflecting focus, persistence, attentiveness, and intelligence, loaded onto Component 3, and it was therefore named Attentiveness. Finally, Component 4 contained traits such as scared, anxious, sensitive, and defensive, and so was called Neuroticism.

### 3.1. Correlations among SR Personality Factor Scales

Pearson’s Correlation showed five significant relationships between the personality subscale scores (Table 3). The strongest of these was between the Attentiveness and Extraversion subscales (r = 0.48, *n* = 805, *p* < 0.01). Attentiveness was also weakly positively correlated with Sociability (r = 0.09, *n* = 823, *p* = 0.01) and Neuroticism (r = 0.10, *n* = 785, *p* = 0.01), while Neuroticism had a slight negative correlation with Extraversion (r = −0.08, *n* = 790, *p* = 0.02) and Sociability (r = −0.09, *n* = 791, *p* = 0.01). With the exception of the Attentiveness and Extraversion correlations, these associations are relatively weak, and it is thus debatable whether or not they are meaningful.

### 3.2. Effects of Ferret Characteristics on SR Personality Dimensions

Within the Sociability dimension, males (2.02 ± 0.76) generally achieved a higher mean subscale score than females (−3.54 ± 0.90); (*F*(1, 646.34) = 22.93, *p* < 0.001). Conversely, within the Attentiveness and Neuroticism dimensions, female ferrets were generally found to be more attentive than males (3.78 ± 1.17 and −3.01 ± 1.02, respectively); *F*(1, 663.07) = 20.75, *p* < 0.001, and also more neurotic than males (2.20 ± 1.04 and −1.67 ± 0.90), respectively; *F*(1, 672.32) = 8.26, *p* < 0.01 (Figure 1). De-sexing was found to have no significant effects on any of the four personality dimensions.

A General Linear Mixed Model analysis with Tukey’s post hoc tests revealed a significant difference among age groups on the Extraversion dimension (*F*(2, 528.91) = 27.673, *p* < 0.001). That is, the level of Extraversion in ferrets shows a general decrease with age (Figure 2).

## 4. Discussion

The main aim of this study was to determine if domestic ferrets have distinct personality types by investigating underlying trends in behavioural traits (personality dimensions). PCA revealed that the underlying components of ferret personality could be described in only 48 words. A four-factor solution emerged from the PCA and accounted for 47.1% of the variance in trait ratings, which, although slightly below the recommended threshold of 50% [35], this value can potentially be improved with further research and refinement of these components.

The four factors were labelled as Extraversion, Sociability, Attentiveness, and Neuroticism, according to the overall behavioural traits that they were comprised. With regards to the influence of ferret characteristics on personality, it was found that sex and age (but not de-sexing) were significant predictors of personality dimensions to some extent.

There is limited research available that specifically investigates personality in mustelids. Those studies that do incorporate elements of personality predominantly use behavioural observations and testing (coding method) rather than trait-rating methodology. Further, the emphasis of this research is mostly on behavioural differences from a reproductive and/or and survival context [36,37], making it challenging to directly compare these results to the current study.

Although not focusing specifically on personality, research comparing “attention response” in domestic ferrets and wild polecats (*Mustela putorius*) demonstrated that, although wild polecats are more attentive than their domesticated counterparts, ferrets nonetheless exhibit a frequent attentive response while investigating an unfamiliar area [38]. This is aligned with our own findings in that it supports the existence of an Attentiveness personality dimension in domestic ferrets.

Personality studies in captive European mink (*Mustela lutreola*) have identified individual differences in boldness and exploratory behaviour [36,39]. One study [36] identified three “personality trait domains” (boldness, exploration, and sociability) based on observations during behavioural testing. Although boldness and exploration did not specifically emerge as personality dimensions in ferrets, the sociability component in mink is still potentially analogous to the Sociability dimension that emerged in the current study. However, behavioural testing of ferrets will be required before making a direct comparison between the two.

Another study investigating exploratory behaviour in captive European mink [39] found that males exhibited a higher level of boldness and exploration compared to females, and females tended to display more of a “flight response”. While all male subjects displayed some level of exploratory behaviour, some females showed none, instead remaining in their nest boxes for the duration of behavioural testing. This was thought to be the result of extreme wariness towards the observer given the higher level of caution shown by female mink compared to males prior to testing. These observations align with our finding that female domestic ferrets are rated as significantly more neurotic than their male counterparts.

### 4.1. A Comparison of Ferret Personality Dimensions with Other Species’ Dimensions

Certain components that emerged in this study, Extraversion and Neuroticism, are some of the most generalised across different species [40], suggesting that the traits that they represent are not as exclusive to humans as it was once believed [41].

The first component in ferrets, Extraversion, appears to incorporate items from two dimensions, Extraversion and Openness, which have emerged individually in several species, such as cats [42], dogs [43], and humans [3]. A combination of these components is not unheard of; a very similar dimension (labelled Playfulness-Curious) was found in both vervet monkeys and snow leopards, which also consisted of elements of both Extraversion and Openness [44,45]. Similar to what has been found in cats and dogs, this dimension represents to what extent a ferret is excitable, playful, and inquisitive. The traits in this dimension have been considered to indicate positive affect and are thus only seen in animals exhibiting suitable welfare [20]. It is, however, important to remember that the absence of these traits does not automatically indicate that welfare is compromised.

Although boldness consistently emerges in other species, the item “bold” was removed from our resulting PCA solution given that it loaded strongly across multiple components (Extraversion and Neuroticism). However, ferrets are known to be generally bold, inquisitive animals [46], and given that they have almost exclusively been bred selectively for hunting, boldness is potentially not as important for ferret personality structure, compared to other, more species-specific traits, i.e., exploratory behaviour and inquisitiveness (useful for hunting).

In ferrets, the personality component, Sociability, consisted of traits such as friendly, calm, and affiliative. In both the human and chimpanzee personality models, this Sociability factor is very similar to the dimension, Agreeableness [47]. Sociability in ferrets is quite similar to what is found in dogs [48], whereas in cats, many of these Sociability traits are instead found across the Impulsiveness and Neuroticism dimensions [20]. Sociability has been studied quite extensively in dogs, and although it is similar to ferrets, some of the Sociability traits have also been identified within the Responsiveness to Training factor [49]. Although there is no Training dimension in ferrets, it is interesting that the trait, obedient, fell within the Sociability dimension.

Attentiveness is a personality dimension consisting of traits such as intelligent, focused, determined, and purposeful. In previous studies, such traits have been dispersed over multiple dimensions. In humans, it is analogous to the Open to Experience dimension [50]; however, in dogs, such traits comprise two different dimensions: Motivation and Training Focus [48]. When making comparisons with other companion animals, it is interesting that those traits relating to training focus in dogs were rated quite differently in ferrets and were thus either eliminated during the extraction process (i.e., trainability) or loaded on completely different dimensions (i.e., obedient loading on the Sociability dimension). This may indicate that, compared to dogs, ferrets are far less focused when it comes to training due to the fact that ferrets, not having lived alongside humans for as long, are far more independent and not as keen to seek the approval of humans. Thus, it appears that ferrets are better compared with cats when it comes to trainability, given that cats do not have a personality dimension relating to such traits either [51,52].

Like Extraversion, the last dimension, Neuroticism, is also related to well-being [20] but is of a negative effect and contains such items as fearful, anxious, and defensive. This component is also widely studied and is found across several species, including humans, dogs, and cats. It is similar to the emotional stability factor found in humans [53] and has an influence on approach and avoidance behaviour in a range of situations as found previously in dogs [54] and cats [20]. Although generally universal, there are slight differences in the traits that are represented by Neuroticism, depending on the species. For instance, the trait, cautious, is found within the Openness to Experience component in humans, whereas in both dogs and ferrets, it is found within the Neuroticism factor and not present at all in cats [55].

### 4.2. Correlations Between Subscales

All ferret personality subscales were significantly related to at least one other subscale; however, some of these associations were quite weak. The strongest positive correlations were between Extraversion and Attentiveness, Extraversion and Neuroticism, and Sociability and Neuroticism. Significant relationships between personality components have also been found in other species such as humans [56], hyenas [4], dogs [48], and cats [57]. This suggests that associations between personality components may be of evolutionary significance, particularly in companion animals [48,57]. Furthermore, it is possible that high ratings on the Attentiveness and Extraversion dimensions are linked with other aspects of behaviour, such as positive environmental interactions. This will require further investigation.

### 4.3. Effects of Ferret Characteristics on SR Personality Dimensions

Female ferrets were rated as overall less sociable than males, which could be related to the finding that they are also more neurotic than males. In turn, higher levels of Neuroticism in females may also be why they are classed as more attentive.

De-sexing did not influence personality in ferrets, which is comparable to what has been found in dogs [48]. However, in another study, de-sexing *was* found to influence calmness in dogs [58]. The variability in these findings could be the result of singular behaviours being the focus of the study instead of personality as a whole [20].

The only significant relationship revealed between personality and age was a general decline with age on the Extraversion dimension. Despite the fact that personality dimensions are believed to be generally resistant to considerable change with time [59], these results were not surprising considering this correlation has also been discovered in humans and other animals, for instance, in dogs [43], especially with relation to gregariousness. This can be attributed to the fact that younger animals engage in more play behaviour than the older age groups [57,60]. Personality would still be developing in individuals within the youngest age group (it is believed that it becomes more consistent with increasing age) [61,62], while individual behavioural differences may be more difficult to detect in ferrets within the oldest age group (due to increased time spent sleeping, higher likelihood of illness, etc.).

### 4.4. Implications and Future Research

This study has revealed four distinct personality dimensions in domestic ferrets, which share several similarities with those dimensions that have emerged in numerous other species. For instance, the two major dimensions, Extraversion and Neuroticism, are thought to be directly related to life-history trade-offs and are largely under the control of physiological, genetic, and cognitive elements [63,64]. The fact that these dimensions also appeared to emerge in ferrets provides further evidence to the notion that personality is adaptively significant.

Now that a framework for personality has been identified in ferrets, we can expand upon this research and implement validation measures to assess its reliability [65]. This species provides an ideal model for this purpose, given the relative ease of sourcing these animals for behavioural observation and assessment. There is also the added benefit that ferrets have only undergone domestication relatively recently (2000–3000 years) and are thus likely to still share certain behavioural characteristics with their wild counterparts (e.g., European polecats, black-footed ferrets, and stoats). This will provide an interesting insight into how personality evolves and which key behavioural traits are maintained despite artificial selection [66,67].

## 5. Conclusions

This research has provided further evidence demonstrating that rating methodology can be used to explore personality in domestic companion animals. Our findings suggest that an underlying personality framework consisting of four dimensions exists in the domestic ferret, which is comparable to what has emerged previously in other species, including humans. It was also revealed that ferret personality differed based on age and sex, with ferret owners generally rating males as more sociable than females, but females as more attentive and neurotic than males. Further, a general decline was observed in extraversion with increasing age of ferrets, indicating younger ferrets tend to be more extraverted. Now that a framework for domestic ferret personality has been established using the rating method, we can attempt to validate these results by subjecting the same sample of ferrets to quantitative behavioural testing, also known as the coding method. This will help determine the extent to which our personality dimensions can predict real and observable traits of individual ferrets.

## Figures and Tables

**Figure 1 animals-11-02173-f001:**
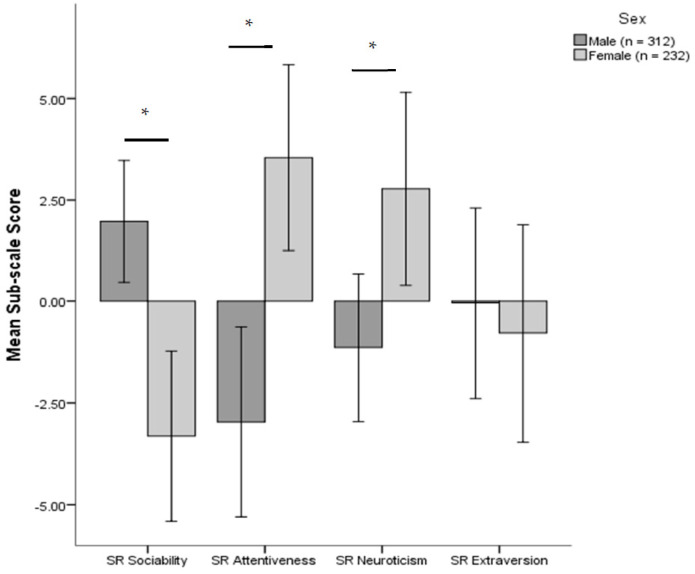
Mean subscale scores for each SR personality dimension, split by sex ± 2 standard errors. * indicates a significant difference between sex at the α = 0.05 level.

**Figure 2 animals-11-02173-f002:**
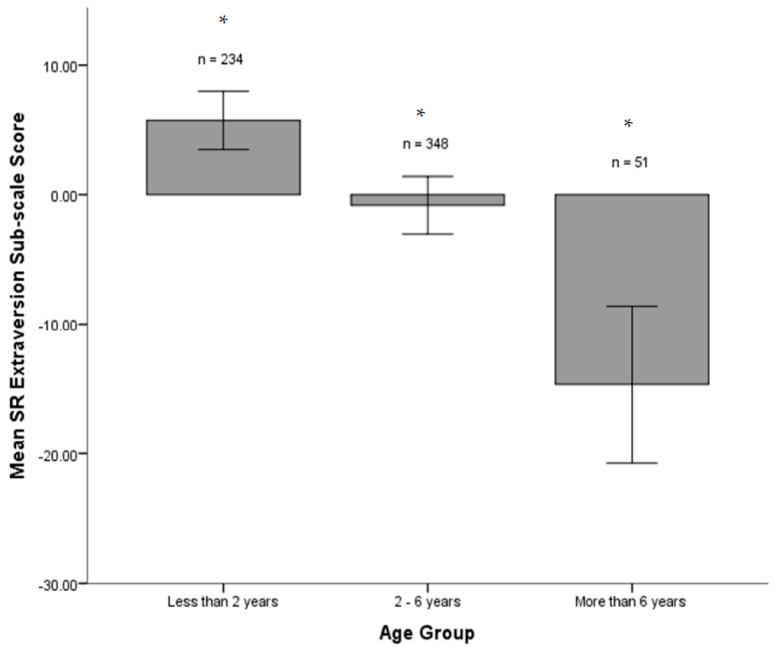
Mean SR Extraversion subscale scores (%) according to age group ± 2 standard errors. * indicates a significant difference at the group *p* < 0.05 level (Tukey’s HSD).

**Table 1 animals-11-02173-t001:** Summary statistics for characteristics of 1029 ferrets.

Characteristic	*n*	Percent
	N	%
Sex		
Male	603	58.6
Female	424	41.2
Missing	2	0.2
De-sexed		
Yes	894	86.9
No	130	12.6
Missing	5	0.5
Age		
Less than 2 years	389	37.8
2–6 years	541	52.6
More than 6 years	92	8.9
Unknown	6	0.6

**Table 2 animals-11-02173-t002:** Principal component analyses results for 1029 ferrets based on owner ratings of 67 traits. Values of 0.4 or above, or −0.4 and below, are bolded, indicating a strong loading for that trait on the respective component.

	Component Number and Label			
	1	2	3	4
	Extraversion	Sociability	Attentiveness	Neuroticism
Eigenvalues	13.061	7.488	4.723	3.570
% of variance explained	16.014	10.916	10.614	9.567
Cronbach’s Alpha	0.911	0.639	0.850	0.696
Traits analysed: Traits retained in model:				
Excitable	**0.768**	−0.073	0.094	0.034
Enthusiastic	**0.764**	0.034	0.228	−0.092
Playful	**0.750**	0.109	0.039	0.028
Zealous	**0.677**	−0.020	0.246	−0.064
Impulsive	**0.677**	−0.200	0.020	0.012
Boisterous	**0.668**	−0.191	0.087	−0.035
Frivolous	**0.630**	0.048	−0.241	−0.029
Hyperactive	**0.619**	−0.275	0.036	0.150
Outgoing	**0.596**	0.326	0.104	−0.303
Eager	**0.590**	−0.040	0.318	−0.068
Extraverted	**0.588**	0.086	0.194	−0.248
Inquisitive	**0.568**	0.025	0.353	−0.192
Exploratory	**0.564**	0.004	0.315	−0.253
Interactive	**0.550**	0.217	0.148	−0.072
Nosey	**0.513**	−0.128	0.189	−0.074
Compulsive	**0.484**	−0.233	0.071	0.072
Interested	**0.479**	0.150	0.378	−0.062
Gentle	−0.105	**0.788**	0.005	−0.132
Friendly	0.188	**0.697**	−0.008	−0.270
Docile	−0.144	**0.688**	0.019	0.009
Tame	0.074	**0.639**	0.091	−0.351
Calm	−0.291	**0.638**	0.143	−0.172
Aggressive	0.007	**−0.591**	0.150	0.170
Relaxed	0.005	**0.577**	0.080	−0.375
Obedient	−0.040	**0.574**	0.204	0.138
Submissive	−0.026	**0.531**	−0.221	0.212
Stubborn	0.278	**−0.471**	0.282	0.018
Affiliative	0.174	**0.437**	0.192	0.005
Perceptive	0.090	0.155	**0.754**	0.054
Intelligent	0.095	0.246	**0.707**	0.012
Focused	0.125	0.012	**0.656**	0.047
Analytical	0.013	0.103	**0.655**	0.103
Purposeful	0.278	−0.068	**0.639**	−0.062
Determined	0.348	−0.246	**0.596**	−0.150
Attentive	0.099	0.174	**0.591**	0.115
Assertive	0.323	−0.252	**0.472**	−0.190
Persistent	0.357	−0.282	**0.468**	−0.037
Independent	0.073	−0.287	**0.465**	−0.138
Alert	0.334	−0.044	**0.445**	0.225
Scared	−0.036	−0.153	−0.077	**0.741**
Wary	−0.048	−0.129	0.112	**0.731**
Fearful	−0.103	−0.041	−0.070	**0.712**
Anxious	−0.071	−0.139	0.004	**0.694**
Skittish	0.027	−0.122	−0.094	**0.694**
Carefree	0.342	0.210	0.128	**−0.571**
Sensitive	0.013	0.179	0.092	**0.537**
Cautious	−0.297	0.230	0.229	**0.494**
Defensive	0.075	−0.294	0.233	**0.473**
Traits removed from model				
Bold	**0.505**	0.330	−0.120	**−0.428**
Sociable	**0.483**	−0.043	**0.406**	−0.076
Hardy	**0.480**	**0.460**	−0.098	−0.338
Vocal	0.395	0.079	0.023	0.148
Strong	0.372	0.316	−0.031	−0.117
Erratic	0.326	−0.096	−0.158	0.106
Trainable	0.057	**0.438**	**0.436**	0.104
Protective	0.051	0.380	0.034	0.249
Versatile	0.341	0.350	0.219	−0.202
Communicative	**0.446**	0.077	**0.460**	−0.267
Consistent	−0.031	0.298	0.329	−0.156
Emotional	0.085	0.304	0.324	0.302
Predictable	−0.064	0.180	0.259	−0.072
High-strung	0.134	0.092	**−0.424**	**0.558**
Confident	0.348	**0.423**	0.022	**−0.546**
Self-assured	0.250	**0.435**	0.079	**−0.533**
Dominant	0.182	**−0.441**	**0.439**	−0.080
Dependent	0.037	0.311	−0.150	0.326
Opportunistic	**0.520**	−0.127	**0.402**	−0.072

**Table 3 animals-11-02173-t003:** Pearson’s correlations between the four personality subscales for ferrets.

		1.	2.	3.	4.	M	S.D
Extraversion	*r* *p* *n*	1.00 834				65.84	20.01
2. Sociability	*r* *p* *n*	0.02 0.64 818	1.00 848			58.02	15.42
3. Attentiveness	*r* *p* *n*	0.48 ** 0.00 805	0.09 ** 0.01 823	1.00 844		70.66	18.88
4. Neuroticism	*r* *p* *n*	−0.08 * 0.02 790	−0.09 * 0.01 791	0.10 ** 0.01 785	1.00 826	32.61	17.15

** *p* < 0.01 level, * *p* < 0.05 level.

## Data Availability

The data presented in this study are available on request from the corresponding author. The data are not publicly available due to Ethics in Human Research restrictions.

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
