# Peer review of "Identifying Behavioural Traits and Underlying Personality Dimensions in Domestic Ferrets (Mustela putorius furo)"

_animals, 2021, doi:10.3390/ani11082173_

Round 1
Reviewer 1 Report
First, I would like to acknowledge the fact that the sample used in this paper is large, which is often difficult to be found in behavioural studies.
Lines 37 to 40, which numerical results support these statements. Numbers must be included to support what is claimed.
Line 40. Add decrease rate. Was it significant or not. P value?
Line 51 change and to or.
Line 51. Change second and to a . and start a new sentence. Like “By making….
When I read the introduction, I feel it is a little bit unfocused on the aims of the paper, and even if only one study working on ferrets can be found in literature it may be better explored. For example, maybe for mustelids there are certain works whose outputs can benefit the interpretation of the result presented here or even may enrich the introduction in regards what the state of the art is at present.
I particularly enjoyed and liked the section describing word selection. I may have added certain stats like interrater reliability as it may somehow support the replicability of raters’ opinions.
Line 131. Remove double dot.
I think still any component loading below 0.5 is low. However, as appointed, sample may enable lowering the limit a little. This must be stated anyway.
The sentence about assumptions must be relocated before explaining that GLMM was used. I have my reservations on all assumptions being fulfilled (according to my background) but if authors claim it to be like that, I do not have anything to say. The important thing is to test it and for test used to match the result after those assumptions, and in this paper it was done or at least claimed to. Maybe adding which test were used for assumption testing and factual results would be necessary to ensure replicability.
I am still impressed for the huge work performed to collect this amount of data. Congratulations.
With Cronbach alpha, it is the same as in component loadings. If sample is large, we can permit values to be slightly lower that the recommended limit of 0.7.
Discussion was good and well written. Still, I missed some references working on ferrets. I know that working on a understudied species can be challenging and will not be the one to take value from discussion because of that. Indeed, this paper is a good resource for future papers. Still authors should add something about personality being evaluated in other mustelids. Try looking in other languages.
Conclusions are too general and do not bring a take home message of the factual results derived from the procedure applied in this study. Please rewrite.
Reviewer 2 Report
Why isn't Reijgwart a co-author
Was social play or object play or both on the survey?
52 Are the functional significances discussed later?
Is accounting for less than half the variation considered sufficient in this field?
In dogs and cats the boldness scale is very robust. Please explain why this is not true of ferrets.
table 2 what does bold mean? Please put in legend
Reviewer 3 Report
As for me, a very interesting methodology and approach to the issue of personality of animals (ferrets). A strong element of the article is the discussion relating to other genres.
I missed the "Material and methods" sample survey form for ferret owner. It is true that examples of individual questions and parts are described, but the whole would be more informative.
In my opinion, it is unnecessary to give the number of sexually unspecified individuals, as well as those of undefined age. These groups are 7 and 6 individuals, respectively.
Round 2
Reviewer 1 Report
All comments have been approached and a valid response has been provided.
Author Response
Please see cover letter (attached)
